# Incidence, Mortality, and Survival Trends in Cancer of the Gallbladder and Extrahepatic Bile Ducts in Lithuania

**DOI:** 10.3390/medicina59040660

**Published:** 2023-03-27

**Authors:** Audrius Dulskas, Dovile Cerkauskaite, Ausvydas Patasius, Giedre Smailyte

**Affiliations:** 1Department of Abdominal and General Surgery and Oncology, National Cancer Institute, 1 Santariskiu Str., LT-08406 Vilnius, Lithuania; 2SMK, University of Applied Social Sciences, LT-08211 Vilnius, Lithuania; 3Institute of Clinical Medicine, Faculty of Medicine, Vilnius University, M. K. Čiurlionio Str. 21/27, LT-03101 Vilnius, Lithuania; 4Faculty of Medicine, Lithuanian University of Health Sciences, A. Mickevičiaus g. 9, LT-44307 Kaunas, Lithuania; 5Department of Public Health, Institute of Health Sciences, Faculty of Medicine, Vilnius University, LT-03101 Vilnius, Lithuania; 6Laboratory of Cancer Epidemiology, National Cancer Institute, 1 Santariskiu Str., LT-08406 Vilnius, Lithuania

**Keywords:** gallbladder, extrahepatic bile ducts, epidemiology, site-specific, incidence, mortality

## Abstract

*Background and Objectives*: Gallbladder cancer is a rare type of cancer, with aggressive clinical behavior. Limited treatment options provide poor survival prognosis. We aimed to investigate the incidence, mortality trends, and survival of gallbladder and extrahepatic bile duct cancer in Lithuania between 1998 and 2017. *Materials and Methods*: The study was based on the Lithuanian Cancer Registry database. The study included all cases of cancer of the gallbladder and extrahepatic bile ducts reported to the Registry in the period 1998–2017. Age-specific and age-standardized incidence rates were calculated. In addition, 95% confidence intervals for APC (Annual Percent Change) were calculated. Changes were considered statistically significant if *p* was <0.05. Relative survival estimates were calculated using period analysis according to the Ederer II method. *Results*: Age-standardized rates for gallbladder cancer and extrahepatic bile duct cancer among females decreased from 3.91 to 1.93 cases per 100.000 individuals between 1998 and 2017, and from 2.32 to 1.59 cases per 100.000 individuals between 1998 and 2017 among males. The highest incidence rates were found in the 85+ group with 27.5/100,000 individuals in females and 26.8/100,000 individuals in males. The 1-year as well as 5-year relative survival rates of both genders were 34.29% (95% CI 32.12–36.48) and 16.29% (95% CI 14.40–18.27), respectively. *Conclusions*: Incidence and mortality from gallbladder and extrahepatic bile duct cancer decreased in both sexes in Lithuania. Incidence and mortality rates were higher in females than in males. Relative 1-year and 5-year survival rates showed a steady increase during the study period among males and females.

## 1. Introduction

According to GLOBOCAN (Global Cancer Incidence, Mortality and Prevalence), gallbladder cancer is relatively rare and stands in the 24th place among the most frequent type of cancers worldwide, with more than 115 949 new cases in 2020 [1]. While being relatively rare, gallbladder cancer is responsible for more than 50% of all biliary tract malignancies [2]. Gallbladder cancer has a tendency to be more common in women. In 2020, the estimated incidence in females was 1.4 per 100,000, while in males, it was 0.9 [1]. The incidence rates are high in Latin America, North Africa, and Asia (in Peru, Chile, Morocco, Egypt, Algeria, or Thailand, the age standardized incidence rates are more than 1.1), are relatively high in some countries in Eastern and Central Europe (i.e., Czech Republic, Slovakia, and Poland), yet low in the United States and most Western and Mediterranean European countries (i.e., UK, France, and Norway) [3].

The underlying causes of the gallbladder cancer are not fully understood; cholelithiasis is thought to be the main risk factor for gallbladder cancer. Gallstone disease causes chronic inflammation which has the strongest correlation with gallbladder malignancy [4]. Gallbladder and bile duct tissues accumulate genetic mutations which can lead to malignancy formation. The most known mutation is in the *TP53 tumor suppressor gene* [5]. Other risk factors include female gender, which is associated with a higher concentration of estrogen causing increased cholesterol super saturation in bile, thus being involved in chronic cholelithiasis and gallbladder cancer [6]. Risk factors for extrahepatic bile duct cancer include congenital cystic changes, primary sclerosing cholangitis, parasitic infections, inflammatory bowel disease, exposure to chemical substances (e.g., thorium dioxide, nitrosamines, polychlorinated biphenyls), several medications (e.g., oral contraceptive pills), and genetic polymorphism [7,8].

In this study we analyzed incidence, mortality trends, and relative survival of gallbladder and extrahepatic bile duct cancer in Lithuania from 1998 to 2016.

## 2. Material and Methods

The study was conducted using the Lithuanian Cancer Registry database covering a population of less than three million residents according to the 2018 census. The Lithuanian Cancer Registry is a population-based cancer registry which contains personal and demographic information (place of residence, sex, date of birth, and vital status), as well as information on diagnosis (cancer site, date of diagnosis, and method of cancer verification) and death (date of death and cause of death) of all cancer patients in Lithuania from 1978. Cancer Registry data are included in Cancer Incidence in Five Continents, where submitted data undergo systematic evaluation of indices of completeness and accuracy [9]. The study included all cases of cancer of the gallbladder and extrahepatic bile ducts reported to the Registry in the period 1998–2017 [ICD-10 (ICD – 10 International Classification of Diseases 10th Revision) codes C23 for gallbladder together with C24 for cancers of extrahepatic bile ducts with ampulla of Vater and other, unspecified malignancies of bile ducts]. All 390 cases identified only by the death certificate (DCO), or only by the autopsy were excluded from survival analysis.

Age-specific and age-standardized incidence rates were calculated. Standardization was performed using the direct method (European standard population, 1976). Corresponding population data, by age, sex and year were available from Statistics Lithuania. The joinpoint regression model was used to provide estimated average annual percentage change (AAPC) and with 0 number of joinpoint allowed, to detect estimated trend over the analyzed period. For each of the identified trends, we also fit a regression line to the natural logarithm of the rates using calendar year as a regression variable. Changes were considered statistically significant if *p* was <0.05. Joinpoint software version 4.3.1.0 (National Cancer Institute, Bethesda, MD, USA) was used. One-year and five-year relative survival estimates were calculated using period analysis. The relative survival was calculated as the ratio of the observed survival of cancer patients and the expected survival of the underlying general population. The latter was calculated according to the Ederer II method, using national life tables for the Lithuanian population stratified by age, gender, and calendar year. All calculations were conducted with STATA 15 (StataCorp LP, College Station, TC, USA); relative survival analysis was performed with the strs module.

## 3. Results

During the 20-year period in Lithuania, 2364 gallbladder cancer cases (782 in males, 1582 in females) were diagnosed and 1317 deaths were registered. The number of gallbladder and extrahepatic bile duct cancer cases decreased from 658 in the period between 1998 and 2002 to 561 cases between 2013 and 2017.

Incidence and mortality of gallbladder and extrahepatic bile duct cancer decreased in both sexes (Figure 1). In females, incidence decreased from 3.91 cases per 100,000 in 1998 to 1.93 cases per 100,000 in 2017 (AAPC 0.02%,). In males, it decreased from 2.32 to 1.59 cases per 100,000 between 1998 and 2017 (AAPC 0.10%) (Figure 2). Incidence of gallbladder and extrahepatic bile duct cancer increases with increasing age (Figure 3).

Mortality rates of gallbladder and extrahepatic bile duct cancer were stable in males and decreased in females. In males, the observed mortality rates were 1.43 in 1998 and 1.58 cases per 100,000 in 2017 (AAPC 0.017%). Female mortality rates significantly decreased from 2.83 to 1.39 cases per 100,000 from 1998 to 2017 (AAPC 2.1%) (Figure 4).

The relative 1-year survival rate of male patients diagnosed with gallbladder and extrahepatic bile duct cancer in the period of 1998–2017 was 41.78% (95% CI 37.80–45.73), and showed a steady increase from 28.25% in the period 1998–2002 to 45.73% in the period 2013–2017. Female patients’ survival rate in the period 1998–2017 was 30.59% (95% CI 28.03–33.18) and survival improved more slowly, from 23.70% in the period 1998–2002 to 38.02% in the period 2013–2017 (Figure 5).

The relative 5-year survival rate of male patients diagnosed with gallbladder and extrahepatic bile duct cancer in the period of 1998–2017 was 20.51% (95% CI 16.86–24.49), and improved over the time from 13.52% in the period 1998–2002 to 22.89% in the period 2013–2017. Female patients’ survival rates in the period 1998–2017 were 14.22% (95% CI 12.12–16.50) and there was a noticeable improvement from 10.65% in the period 1998–2002 to 19.82% in the period 2013–2017 (Figure 6).

## 4. Discussion

This study has presented for the first time the incidence, mortality, and survival rates of gallbladder and extrahepatic bile duct cancer in Lithuania from 1998 to 2017. The decrease in incidence and mortality and improved survival rates were seen among both sexes during study period.

In Lithuania, the number of annual new cases of gallbladder and extrahepatic bile duct cancer gradually decreased. The highest incidence was found in 1998 with 3.91 cases per 100,000 individuals in females and 2.32 cases per 100,000 individuals in males. The same trend was seen in Scandinavian countries, including Denmark, Finland, Norway, and Sweden, where the highest incidence was found in the 1980s in Sweden with five cases per 100,000 individuals in females and three cases per 100,000 individuals in males (World Standard Population) [10]. Hemminki et al. hypothesized that Thorotrast, a suspension of radioactive thorium dioxide, which was used as a radiocontrast agent until the 1950s, is responsible for the higher incidence of gallbladder and extrahepatic bile duct cancer in the 20th century than in the 21st century [11]. Studies from Sweden and the United States confirmed this hypothesis [11,12,13]. The declining pattern was also seen in the number of countries in South and North America, including Brazil, Chile, Colombia, Ecuador, Canada, and the United States [14]. The highest incidence rates of gallbladder and extrahepatic bile duct cancer were found in Chile and Ecuador, probably due to the higher incidence of gallstones and salmonella infection, which are the risk factors for gallbladder cancer [15].

Some genes have a role in gallbladder and extrahepatic bile duct cancer carcinogenesis, including *TP53* and *CDKN2*. Several studies showed a relationship between p53 protein overexpression and gallbladder cancer pathogenesis [16,17,18]. The overexpression of the p53 protein can be found in more than 50% of gallbladder cancer cases [19]. In addition to the *TP53 gene*, mutations of the *KRAS/NRAS* and *IDH1/2 genes* were found in extrahepatic bile duct cancer patients. While genetic factors play a role in gallbladder cancer and extrahepatic bile duct cancer pathogenesis, some different etiological factors have an impact too. One of the most important factors is a chronic inflammation of the gallbladder caused by gallstone disease [20]. Carcinogenesis of gallbladder cancer occurs mainly through a metaplasia–dysplasia–carcinoma pathway. In addition, bacterial infection with a Salmonella Typhi has an impact on gallbladder cancer carcinogenesis as well [20,21,22,23]. A strong association exists between a liver fluke and bile duct cancer [24,25]. Mechanical injury from migrating flukes, and flukes’ metabolites induce tissue damage, induce cell proliferation, and hyperplasia of biliary epithelial cells. As a result, chronic inflammation, and finally bile duct cancer occurs [24]. Other established risk factors include porcelain gallbladder and congenital biliary cysts [15].

The incidence rates of gallbladder and bile duct cancer increases with age. In the Lithuanian population, the highest incidence rates were found in the 85+ group, with 27.52 cases per 100,000 individuals in females and 26.8 cases per 100,000 individuals in males. It could be possibly explained by a cumulative effect of tobacco smoking, inflammatory bowel disease, primary sclerosing cholangitis, gallstones, obesity, cirrhosis, and hepatitis B and C [26]. Hence, the prevalence of the previously mentioned risk factors in Lithuania is unknown. Similar trends were reported in other studies from the United Kingdom and United States [27].

In Lithuania, the incidence and mortality rates were higher in females than in males. Similar sex differences in gallbladder and bile duct cancer were reported in other countries [19,28,29,30].

It was shown that the incidence rates of gallbladder cancer in the United States were found to be higher among females, up to six times more than among males, due to the increased super saturation of cholesterol in bile by estrogens [31,32]. Thus, the incidence of cholelithiasis in females is much higher and it is believed to be one of the reasons for the greater risk of gallbladder cancer carcinogenesis in women [6,30,33,34]. In contrast, extrahepatic bile duct cancer is more commonly diagnosed in males than in females worldwide [14]. Due to this type of cancer being relatively rare in our population, we did not calculate its statistics separately from gallbladder cancer.

The relative 1-year as well as 5-year survival in patients with gallbladder and extrahepatic bile duct cancer increased in the Lithuanian population from 24.99% to 41.33% and from 11.45% to 21.22%, respectively. Similar survival rates were reported by other authors. For instance, Lepage et al. found that 5-year relative survival rates increased from 4.5% to 6.7% in a 30-year period in France [29]. Fong et al., Dixon et al., as well as Konstantinidis et al. from the United States, discovered similar 5-year survival rates, which were 38%, 35% and 35–46%, respectively, in gallbladder cancer patients [35,36,37]. Bjerregaard et al. found a minor improvement in 5-year relative survival from 6–9% to 13–16% from 1980 to 2012 in Denmark’s population [28]. Kang et al. found an increase in 5-year survival rates from extrahepatic bile duct cancer, from 23.1% to 27.5% (*p* < 0.01) in the period between 1999 and 2019 in the Korean population [38]. While the overall 5-year relative survival rates are similar worldwide, the survival rates can differ depending on the stage of disease as well as the therapy that a patient received. According to Marcano-Bonilla et al., the 5-year relative survival in patients with stage 0 gallbladder cancer disease can be 80%, while with stage IVB gallbladder cancer disease, it can be only 2% [19]. The apparent increase in survival throughout the world may be related to more accurate diagnostic tools and improved and increased availability of treatment options. Despite that, gallbladder tract cancers, including gallbladder cancer and extrahepatic bile duct cancer, are highly lethal due to its aggressiveness as well as it being therapy resistant. Another theoretical explanation of the decrease in incidence and increase in survival is the increased numbers of laparoscopic cholecystectomy. The increase in the rate of elective cholecystectomy that occurred following the introduction of laparoscopic cholecystectomy in 1991 was associated with an overall reduction in the incidence of severe gallstone disease, which was entirely attributable to a reduction in the incidence of acute cholecystitis, possibly leading to the decrease in cancer too [39]. Laparoscopic cholecystectomy appears to be one of the surgical procedures performed most often worldwide. However, there is no data to support this relation.

The major strength of our study is its population-based design and complete inclusion of all citizens diagnosed with gallbladder and extrahepatic bile duct cancer.

Our study obviously has some limitations. Firstly, the descriptive design using ecological data, which was not recorded for this specific research objective, increases the risk of information bias. The fact that both incidence and mortality decreased supports a non-artefactual effect. Lack of information on comorbidities, diet, drug use, and other prevalent risk factors limits the opportunity for targeted analysis of association for risk factors with incidence, mortality, and survival data.

In conclusion, incidence and mortality from gallbladder and extrahepatic bile duct cancer decreased in both sexes in Lithuania. Incidence and mortality rates were higher in females than in males. Relative 1-year and 5-year survival rates showed a steady increase during the study period among males and females.

## Figures and Tables

**Figure 1 medicina-59-00660-f001:**
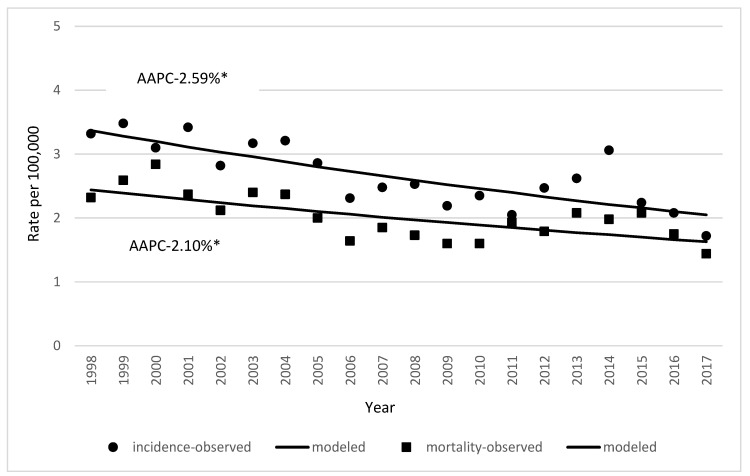
Age-standardized incidence and mortality rates of gallbladder and extrahepatic bile duct cancer in Lithuania in the period 1998–2017. Both sexes. (* indicates that AAPC is significantly different from zero at alpha = 0.05).

**Figure 2 medicina-59-00660-f002:**
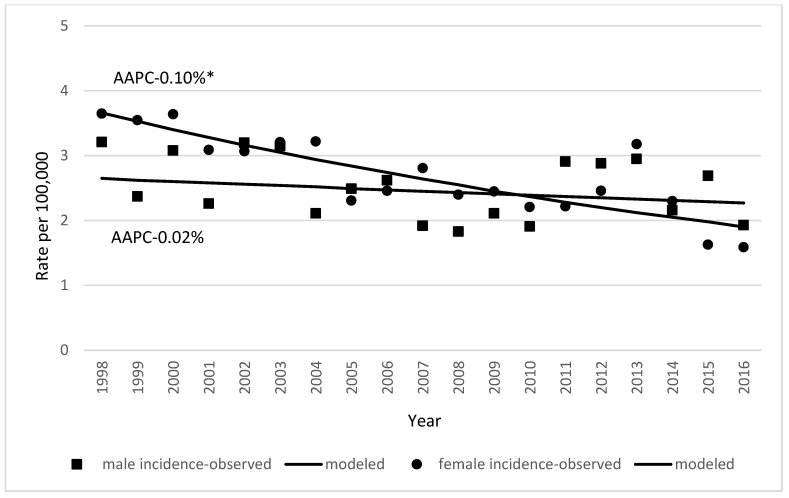
Age-standardized incidence of gallbladder and extrahepatic bile duct cancer by sex in Lithuania in the period 1998–2017 (* indicates that AAPC is significantly different from zero at alpha = 0.05).

**Figure 3 medicina-59-00660-f003:**
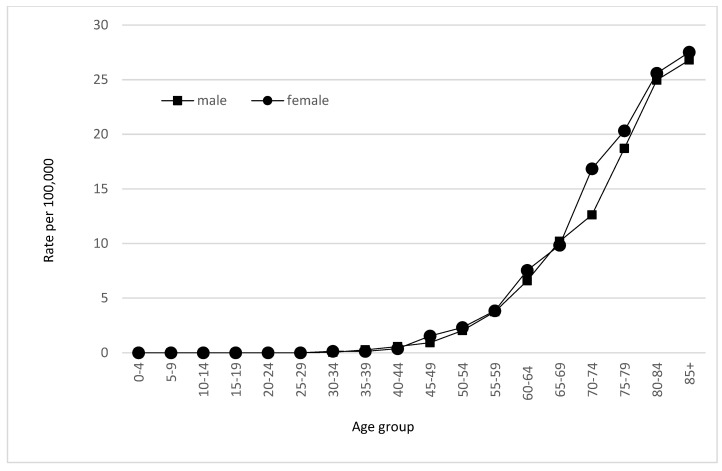
Age-specific incidence of gallbladder and extrahepatic bile duct cancer by sex in Lithuania, 1998–2017.

**Figure 4 medicina-59-00660-f004:**
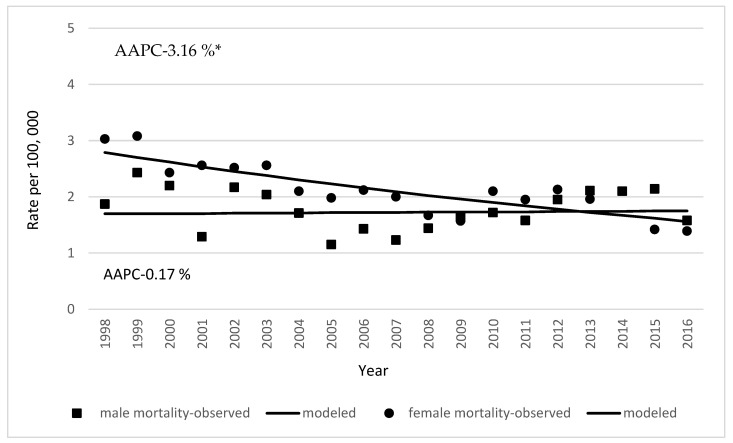
Age-standardized mortality rates of gallbladder and extrahepatic bile duct cancer by sex in Lithuania in the period 1998–2017. (* indicates that AAPC is significantly different from zero at alpha = 0.05).

**Figure 5 medicina-59-00660-f005:**
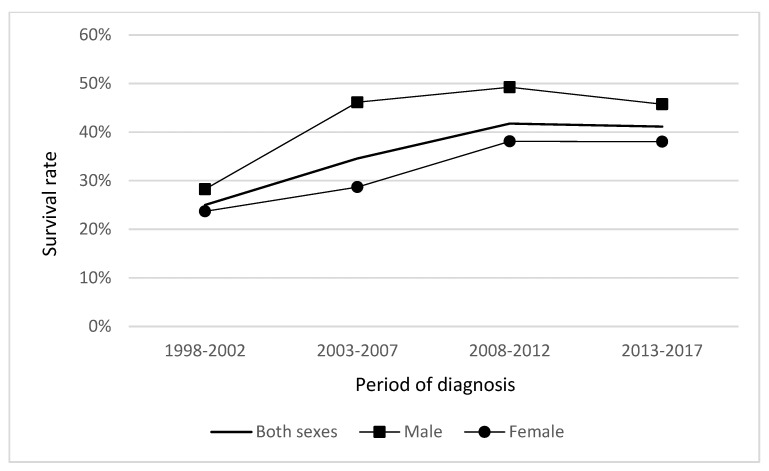
Relative 1-year survival of gallbladder and extrahepatic bile duct cancer patients in Lithuania by period of diagnosis.

**Figure 6 medicina-59-00660-f006:**
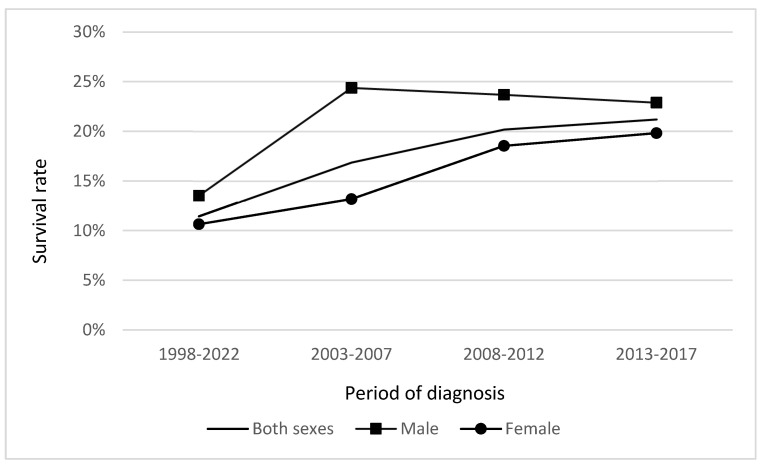
Relative 5-year survival of gallbladder and extrahepatic bile duct cancer patients in Lithuania by period of diagnosis survival.

## Data Availability

The datasets used and/or analyzed during the current study are available from the corresponding author on reasonable request.

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
