# Peer review of "Incidence, Mortality, and Survival Trends in Cancer of the Gallbladder and Extrahepatic Bile Ducts in Lithuania"

_medicina, 2023, doi:10.3390/medicina59040660_

Round 1
Reviewer 1 Report
The submitted paper is original article entitled as Incidence, mortality and survival trend in cancer of the gallbladder and extrahepatic bile duct in Lithuania.
This study was made based on the Lithuanian cancer registry database. Therefore, the quality of database or reliability of the database is important for the readers. Although it is not the popular database such as SEER or NCI, you should introduce the database. For example, national database or not, the starting date, the number of enrolled cancer patients, or which cancers, and so on. Especially, you should describe the detail of the information that database include. In this manuscript, there are only three factors, i.e., incidence, survival and mortality. There was no demographics of the patients or stage, or treatment and so on.
I recommend that you should check the database and add the data of population characteristics. If there was no other data that you describe already, then you should mention clearly which information is included in the Lithuanian cancer registry database and you can analyze only the three factors.
I cannot find figures in the submitted file. Please insert the figures in the manuscript.
Author Response
Answers to Reviewers:
Reviewer 1:
The submitted paper is original article entitled as Incidence, mortality and survival trend in cancer of the gallbladder and extrahepatic bile duct in Lithuania.
This study was made based on the Lithuanian cancer registry database. Therefore, the quality of database or reliability of the database is important for the readers. Although it is not the popular database such as SEER or NCI, you should introduce the database. For example, national database or not, the starting date, the number of enrolled cancer patients, or which cancers, and so on. Especially, you should describe the detail of the information that database include. In this manuscript, there are only three factors, i.e., incidence, survival and mortality. There was no demographics of the patients or stage, or treatment and so on.
Thank you for your prompt review and response regarding our manuscript on Incidence, mortality and survival trend in cancer of the gallbladder and extrahepatic bile duct in Lithuania.
We added short introduction about completeness and quality control of Lithuanian Cancer Registry dataset in Materials and methods section of our manuscript. Unfortunately, due to high number of unknown stage of the disease and lack of information about treatment modalities for particular site of cancer.
More information about demography of the patients is presented in supplementary table as it was proposed by reviewer 2.
I recommend that you should check the database and add the data of population characteristics. If there was no other data that you describe already, then you should mention clearly which information is included in the Lithuanian cancer registry database and you can analyze only the three factors.
All the data collected by cancer registry is listed in the Materials and methods section.
I cannot find figures in the submitted file. Please insert the figures in the manuscript.
We sincerely apologize for this misunderstanding. Figures were submitted in a separate file according to MPDI author guidelines. Now you can find figures in the updated version of manuscript.

Reviewer 2 Report
This manuscript (especially the sections Results, Discussion with paragraph about the strength and limitations of the study, as well as conclusions) is not written in a quality manner, with the presence of numerous mistakes and shortcomings. Some of the mistakes and shortcomings: Line 19: Should it be stated `1998-2017` instead of` 1998-2016`? Line 31: Correct the Keywords. Line 57: Check the Abstract (Line 17, Line 19, Line 24, Line 25) to match data with the period specified in the goals of this manuscript. Line 76: Give an explanation why only `... 0 Number of Joinpoint Allowed, ...`. The choice of allowing 3 joinpoints would have been better, particularly taking into consideration that data for only one country are presented. Line 88: In the entire text of the section Results authors MUST describe and specify the trends displayed on Figures 1, 3 and 4, ie which changes were statistically significant and which trends were not statistically significant. Figures are incorrectly cited in the text: check whether all figures are cited in the appropriate place in the text and correct them. Additionally, pay attention to Figure 2, and clarify what is actually shown on it: if the trends in 1998-2017 were presented, then enter AAPC; if they are not trends but a cross section at the annual level, then accordingly correct the title of this Figure and the description in the text. Lines 89-92: Display all the data listed in this paragraph on one Table. In addition to the number of new cases and deaths by year from 1998 to 2017, show the corresponding age-standardized rates both for incidence and mortality. Lines 95-96: This description does not match Figure 2. Correct. Lines 96-97: Check and correct. Describe accurately and in detail. Lines 98-101: Completely incorrect description of Figure 4. Line 134: Should the reference number in that sentence be corrected? Lines 135-150: In order for that paragraph to justify its role in this Discussion, provide data from the literature on the frequency of those risk factors in Lithuania and other countries, which would be a possible explanation for the described differences in incidence and mortality rates between different countries (with citations of the relevant references). Lines 151-154: Give a possible explanation for the distribution of gallbladder and extrahepatic bile ducts cancer incidence by age, citing appropriate references. Lines 186-191: Cited references No 37 and No 38 do not exist in the list of References. Lines 192-196: Unnecessary repetition of what is presented in this paper will not indicate the strength of this study. Line 196: Specify quality indicators for the population Cancer Registry data of Lithuania. Lines 197-199: Do these sentences state the advantages or limitations of this study? Line 200: Discuss the issue of data quality for both incidence and mortality of gallbladder and extrahepatic bile ducts cancer in the population Cancer Registry data of Lithuania. Line 200: Explain why mortality trends in 1998-2017 by AGE were not presented. Lines 201-202: The conclusion of the paper did not reflect the most important results.
Author Response
Thank you for your prompt review and response regarding our manuscript on Incidence, mortality and survival trend in cancer of the gallbladder and extrahepatic bile duct in Lithuania.
Reviewer 2
This manuscript (especially the sections Results, Discussion with paragraph about the strength and limitations of the study, as well as conclusions) is not written in a quality manner, with the presence of numerous mistakes and shortcomings. Some of the mistakes and shortcomings: Line 19: Should it be stated `1998-2017` instead of` 1998-2016`? Line 31: Correct the Keywords. Line 57: Check the Abstract (Line 17, Line 19, Line 24, Line 25) to match data with the period specified in the goals of this manuscript.
Thank You for your noticing of mistakes. Corrections were made according to your suggestions, and you can find them in the updated version of the manuscript.
Line 76: Give an explanation why only `... 0 Number of Joinpoint Allowed, ...`. The choice of allowing 3 joinpoints would have been better, particularly taking into consideration that data for only one country are presented.
Thank you for your question. The rationale for presenting trends with 0 joinpoints is to provide a clear and concise summary of the data about incidence and mortality of gallbladder and extrahepatic cancer without adding unnecessary complexity. It allows the audience to quickly understand the trend and make informed decisions based on the data presented.
Line 88: In the entire text of the section Results authors MUST describe and specify the trends displayed on Figures 1, 3 and 4, ie which changes were statistically significant and which trends were not statistically significant. Figures are incorrectly cited in the text: check whether all figures are cited in the appropriate place in the text and correct them. Additionally, pay attention to Figure 2, and clarify what is actually shown on it: if the trends in 1998-2017 were presented, then enter AAPC; if they are not trends but a cross section at the annual level, then accordingly correct the title of this Figure and the description in the text.
Thank you for noticing our mistake. Corrections are made according to your suggestions. Statistical significance is shown in the explanation of figures.
Lines 89-92: Display all the data listed in this paragraph on one Table. In addition to the number of new cases and deaths by year from 1998 to 2017, show the corresponding age-standardized rates both for incidence and mortality.
The descriptive table according to your suggestion is added to the manuscript.
Lines 95-96: This description does not match Figure 2.Correct. Lines 96-97: Check and correct. Describe accurately and in detail. Lines 98-101: Completely incorrect description of Figure 4. Line 134: Should the reference number in that sentence be corrected?
Thank you for noticing our mistake. Corrections are made according to your suggestions.
Lines 135-150: In order for that paragraph to justify its role in this Discussion, provide data from the literature on the frequency of those risk factors in Lithuania and other countries, which would be a possible explanation for the described differences in incidence and mortality rates between different countries (with citations of the relevant references).
Thank you for your questions. Unfortunately, there is no reliable information source on prevalence of risk factors of gallbladder and extrahepatic cancer in Lithuania and other countries.
Lines 151-154: Give a possible explanation for the distribution of gallbladder and extrahepatic bile ducts cancer incidence by age, citing appropriate references.
Thank you for your comment, we added possible explanation according to your suggestions.
Lines 186-191: Cited references No 37 and No 38 do not exist in the list of References.
Thank you for your comment, we corrected the list of references.
Lines 192-196: Unnecessary repetition of what is presented in this paper will not indicate the strength of this study. Lines 197-199: Do these sentences state the advantages or limitations of this study?
Thank you for noticing that. You can find strengths and limitatons of our study in updated version of our manuscript.
Line 196: Specify quality indicators for the population Cancer Registry data of Lithuania.
Thank you for your suggestion. It is mentioned in Materials and methods section.
Line 200: Discuss the issue of data quality for both incidence and mortality of gallbladder and extrahepatic bile ducts cancer in the population Cancer Registry data of Lithuania.
This question is partly covered in limitations section of our manuscript.
Line 200: Explain why mortality trends in 1998-2017 by AGE were not presented.
We havent presented age-specific mortality in the manuscript due to almost same pattern of the results as age-specific incidence. As it would not provide additional scientific value, authors decided not to make manuscript overladen.
Lines 201-202: The conclusion of the paper did not reflect the most important results.
Thank you for your comments. Now you can find corrected conclusions in the updated version of manuscript

Round 2
Reviewer 2 Report
Thank you for the opportunity to re-review the manuscript ID: medicina-2156124. The authors have addressed majority of the issues highlighted in my review and satisfactorily responded to my questions. Also, the authors gave some acceptable explanations for the comments. The authors made the necessary changes to the manuscript. I believe that the changes they have made have significantly improved the manuscript. The revised manuscript is clear, readable and informative and will provide a valuable findings for this issue. Thank you to the authors for their responses to my comments.